# Multidrug-Resistant Healthcare-Associated Infections in Neonates with Severe Respiratory Failure and the Impacts of Inappropriate Initial Antibiotic Therap

**DOI:** 10.3390/antibiotics10040459

**Published:** 2021-04-18

**Authors:** Jen-Fu Hsu, Shih-Ming Chu, Hsiao-Chin Wang, Chen-Chu Liao, Mei-Yin Lai, Hsuan-Rong Huang, Ming-Chou Chiang, Ren-Huei Fu, Ming-Horng Tsai

**Affiliations:** 1Division of Pediatric Neonatology, Department of Pediatrics, Chang Gung Memorial Hospital, Taoyuan 333, Taiwan; hsujanfu@cgmh.org.tw (J.-F.H.); kz6479@cgmh.org.tw (S.-M.C.); ccliao@cgmh.org.tw (C.-C.L.); a9275@cgmh.org.tw (M.-Y.L.); qbonbon@cgmh.org.tw (H.-R.H.); cmc123@cgmh.org.tw (M.-C.C.); rkenny@cgmh.org.tw (R.-H.F.); 2College of Medicine, Chang Gung University, Taoyuan 333, Taiwan; b9605032@cgmh.org.tw; 3Division of Pediatric Pulmonology, Department of Pediatrics, Chang Gung Memorial Hospital, Taoyuan 333, Taiwan; 4Division of Neonatology and Pediatric Hematology/Oncology, Department of Pediatrics, Chang Gung Memorial Hospital, Yunlin 638, Taiwan

**Keywords:** ventilator-associated pneumonia, respiratory failure, neonates, multidrug-resistant pathogens, broad-spectrum antibiotics

## Abstract

Background: Multidrug-resistant (MDR) pathogens have emerged as an important issue in neonatal intensive care units (NICUs), especially in critically ill neonates with severe respiratory failure. We aimed to investigate neonatal healthcare-associated infections (HAIs) caused by MDR pathogens and the impacts of inappropriate initial antibiotic therapy on the outcomes. Methods: We retrospectively analyzed all cases of HAIs in neonates with severe respiratory failure in a tertiary-level NICU in Taiwan between January 2014 and May 2020. All clinical features, microbiology, therapeutic interventions, and outcomes were compared between the MDR-HAI and non-MDR HAI groups. Multivariate regression analyses were used to investigate independent risk factors for sepsis-attributable mortality. Results: A total of 275 critically ill neonates with severe respiratory failure who had HAIs were enrolled. Ninety-five cases (34.5%) were caused by MDR pathogens, and 141 (51.3%) cases had positive bacterial cultures from multiple sterile sites. In this cohort, the MDR-HAI group was more likely to receive inappropriate initial antibiotic therapy (51.0% versus 4.7%, respectively; *p <* 0.001) and exhibit delayed control of the infectious focus (52.6% versus 37.8%, respectively; *p* = 0.021) compared with the non-MDR HAI group. The sepsis-attributable and final in-hospital rates were 21.8% and 37.1%, respectively, and they were comparable between the MDR-HAI and non-MDR HAI groups. Empirically broad-spectrum antibiotics were prescribed in 76.7% of cases, and inappropriate initial antibiotic treatment was not significantly associated with worse outcomes. Independent risk factors for sepsis-attributable mortality in neonates with severe respiratory failure included the presence of septic shock (OR: 3.61; 95% CI: 1.54–8.46; *p* = 0.003), higher illness severity (OR: 1.33; 95% CI: 1.04–1.72; *p* = 0.026), and neonates with bronchopulmonary dysplasia (OR: 2.99; 95% CI: 1.47–6.09; *p* = 0.003). Conclusions: MDR pathogens accounted for 34.5% of all neonatal HAIs in the NICU, but neither MDR pathogens nor inappropriate initial antibiotics were associated with final adverse outcomes. Because the overuse of broad-spectrum antibiotics has emerged as an important issue in critically ill neonates, the implementation of antimicrobial stewardship to promote the appropriate use of antimicrobials is urgently needed.

## 1. Introduction

Neonatal sepsis is the most common healthcare-associated infection (HAI) in neonatal intensive care units (NICUs) and is associated with significantly increased morbidity and in-hospital mortality [1,2]. Approximately 20–25% of NICU patients experience at least one episode of neonatal sepsis during hospitalization, and the proportion is particularly high in extremely preterm neonates or those with underlying chronic comorbidities [1,2,3,4]. The population-level estimate of neonatal sepsis was reported to be 2202 (95% CI: 1099–4360) per 100,000 live births, with a mortality rate between 11% and 19% [3]. Neonatal sepsis is the most common cause of antibiotic prescription in NICUs [5]. *Escherichia coli* and *Streptococcus agalactiae* (Group B strep (GBS)) are the most common pathogens of early-onset sepsis, while coagulase-negative *Staphylococcus* (CoNS) accounts for nearly half of neonatal late-onset sepsis [6,7,8].

Neonatal sepsis occasionally occurs in extremely preterm infants with underlying pulmonary comorbidities or in critical neonates with respiratory failure [3,4,9]. In these situations, clinicians tend to use empiric broad-spectrum antibiotics because they cannot take the risk of clinical deterioration [10,11]. In addition, multidrug-resistant (MDR) pathogens are more likely to be associated with a higher severity of illness and a higher mortality rate, especially when the neonates do not receive adequate antibiotic treatment on time [12,13]. In some cases, these patients experience culture-negative sepsis due to previous exposure to broad-spectrum antibiotics, which potentially cause emergence of MDR pathogens [14,15,16]. Though the use of broad-spectrum antibiotics in neonates with a high risk of meningitis is supported by both Infectious Diseases Society of America and the American Academy of Pediatrics [17,18], the impact of inappropriate initial antibiotic administration on septic neonates with severe respiratory failure has not been documented. We hypothesized that neonatal sepsis combined with severe respiratory failure is challenging and aimed to examine the impacts of empiric antibiotic administration and MDR pathogens on the outcomes.

## 2. Methods

### 2.1. Patients, Study Design, and Setting

All neonates with severe respiratory failure who had bacterial sepsis in the NICUs of Chang Gung Memorial Hospital (CGMH) between January 2014 and May 2020 were reviewed, and their records were retrieved for analyses. The NICUs of CGMH contain a total of three units, with a total capacity of 49 beds equipped with ventilators and 58 beds of special care nurseries. The CGMH is the largest tertiary-level medical center in Taiwan, and more than one-fourth of all premature infants and critically ill neonates are hospitalized in the NICUs of CGMH. All neonatal early-onset sepsis, late-onset sepsis, meningitis, and sterile site culture positive nosocomial infections were enrolled, including ventilator-associated pneumonia, catheter-related bloodstream infections (BSIs), and intra-abdominal infections [19]. All these nosocomial HAIs were confirmed and diagnosed based on the standard criteria of the Centers for Disease Control and Prevention (CDC) [19,20]. This study was approved by the Institutional Review Board of CGMH (the certificate no. is 201700988A3), and written informed consent was waived because all patient records/information were anonymized and deidentified prior to analysis.

### 2.2. Definitions of HAIs, Severe Respiratory Failure, and Enrolled Criteria

For various HAIs in this study, the updated diagnostic criteria of the CDC were applied. We retrospectively reviewed all radiological, clinical, and microbiological data of neonatal ventilator-associated pneumonia (VAP) cases, and those fulfilling the strict diagnostic criteria of VAP in the NICU (Appendix A
Table A1) were enrolled. The definition of repeated or recurrent HAI isolates was based on previous studies [20,21]. The definition of CoNS late-onset sepsis was based on our previous publications [22]. In each patient with severe respiratory failure, only the first episode of HAI was enrolled for analyses. Severe neonatal respiratory failure is defined by at least one of the following: (1) a requirement of inspired oxygenation fraction (FiO_2_) ≥ 50% for more than 24 h; (2) an oxygenation index (OI) ≥ 20; (3) AaDO_2_ ≥ 400; and (4) a requirement of more than two cardiac inotropic agents (usually dopamine and dobutamine), with at least one of them ≥10 ug/kg/min to maintain adequate blood pressure.

The following first-line antibiotics, namely ampicillin/sulbactam, oxacillin, ceftriaxone, and gentamicin, were defined as limited-spectrum antibiotic therapy [23]. In each HAI episode, we considered all positive bacterial isolates from sterile sites, including blood, cerebrospinal fluid, pleural effusion, ascites, urine, catheter tip (excluding contamination), and tracheal aspirates (excluding colonization). Resistance was defined if one of the isolated bacterial strains was resistant to the prescribed antibiotics. Inappropriate initial antibiotic treatment was defined when one of the isolated bacteria strains in the HAI cases was resistant to the empirical treatment. The National Committee for Clinical Laboratory Standards Institute (CLSI) has guidelines for the disk diffusion method [24], and categorical assignment was performed using CLSI breakpoints, which were used to determine the antibiotic susceptibility patterns for all isolated microorganisms [24]. The bacterial strain was considered an MDR pathogen if it was resistant to at least one agent in ≥3 of the following antimicrobial categories: broad-spectrum cephalosporins (ceftazidime and cefepime), carbapenem, aminoglycosides, penicillins (including piperacillin/tazobactam), monobactams (aztreonam), and fluoroquinolones [12,25]. When one of the isolated bacterial strains in polymicrobial HAIs was resistant to ≥3 antimicrobial categories, it was also considered as an episode of MDR HAI.

We applied the latest updated diagnostic criteria in the standard textbook of neonatology [26] to define all diagnostic entities of prematurity, including respiratory distress syndrome (RDS), the grading of intraventricular hemorrhage (IVH), and the stages of bronchopulmonary dysplasia (BPD) or persistent pulmonary hypertension of the newborn (PPHN). The presence of various comorbidities was considered only at the onset and therapeutic course of the HAI cases.

### 2.3. Data Collection

Patient demographics, clinical features, laboratory data, therapeutic interventions including the modification of antibiotics, and outcomes were retrospectively reviewed and collected using standard forms for all the HAI cases. The onset of the HAI episode was defined as the sampling of the first positive bacterial cultures from the presumed sterile sites. We used the neonatal therapeutic intervention scoring system (NTISS) [27] to evaluate the illness severity at the onset of each HAI episode. The oxygenation index (OI) was also calculated based on the standard definition [28]. Patients who were transferred to other hospital or had unknown outcomes due to poorly kept records were excluded from the analyses.

### 2.4. Statistical Analysis

We used the mean (standard deviation (SD)) to present variables with parametric distributions and the median (interquartile range (IQR)) to present data with nonparametric distributions. Student’s *t*-test and the Wilcoxon rank sum tests were used for comparisons between continuous variables of different subgroups, while categorical variables were compared using chi-square tests or Fisher’s exact tests. All *p*-values were two tailed, and *p*-values < 0.05 were considered to be statistically significant.

We investigated the impacts of MDR pathogens and inappropriate antibiotic therapy on the outcomes, which included sepsis-attributable mortality, final in-hospital mortality and response to antibiotic treatment. The primary outcome was HAI-attributable mortality, and the secondary outcome was final in-hospital mortality. HAI-attributable mortality was defined as death within 3 days of the HAI onset, persistent bacteremia or infectious complications until mortality, or those with progressive deterioration until mortality. We used univariate and multivariate logistic regression analyses to investigate the independent risk factors for sepsis-attributable mortality. All variables with *p*-values < 0.1 were enrolled in the multivariate logistic regression model. All statistical analyses were performed using SPSS (version 21.0; IBM, Armonk, NY, USA).

## 3. RESULTS

### 3.1. Epidemiology of HAIs in Neonates with Severe Respiratory Failure

During the study period, a total of 275 cases of neonatal HAIs in critically ill neonates with severe respiratory failure were identified and enrolled for analyses. Among these cases, the most common HAIs were bloodstream infections (41.1%; *n* = 113), followed by VAP (32.0%; *n* = 88), catheter-related BSI (12.7%; *n* = 35), intra-abdominal infection (5.8%; *n* = 16), urinary tract infection (2.2%; *n* = 6), and meningitis (1.5%; *n* = 4). The last 13 cases of HAIs had combined VAP and intra-abdominal infections. Among the 275 HAI cases, blood cultures were positive in 252 (91.6%) cases, and 141 (51.3%) cases had positive bacterial cultures from two or more sterile sites (Table 1). The median (IQR) gestational age (GA) and birth weight of this cohort were 26.0 (25.0–29.0) weeks and 838 (721.0–1080.0) g, respectively. The HAIs occurred at 25.0 (12.0–56.0) (median (IQR)) days of life, 85.5% of all neonates had a very low birth weight (BBW < 1500g), and 65.5% of the neonates had a GA < 28 weeks.

The causative pathogens of all the HAIs are presented in Table 1. In 275 cases of HAIs, a total of 450 strains of microorganisms were identified as pathogens, including 253 (56.2%) Gram-negative bacilli, 173 (38.4%) Gram-positive cocci, and 24 fungal species. The most common pathogenic microorganism found in the HAIs was *Staphylococcus aureus* (85 (18.9%)), followed by CoNS (64 (14.2%)), *E coli* (62 (13.8%)), *K. pneumonia* (51 (11.3%)), *Acinetobacter baumannii* (33 (7.3%)), and *Pseudomonas aeruginosa* (31 (6.9%)). Among the 275 cases of HAIs, 95 (34.5%) were caused by MDR pathogens, and the most common MDR pathogens were methicillin-resistant *Staphylococcus aureus* (MRSA) (42 (9.3%)), extended-spectrum β-lactamase (ESBL)-producing Gram-negative bacilli (31 (6.9%)), and *Stenotrophomonas maltophilia* (22 (4.9%)). Nearly half of these HAI cases (46.5%; *n* = 128) were caused by more than one microorganism, and 19 (6.9%) cases had combined fungal infections.

### 3.2. Comparisons between MDR HAIs and Non-MDR HAIs

The patient demographics, clinical features, chronic comorbidities, and infectious focuses of all HAI cases are presented in Table 2. Based on the NTISS score and OI data at the onset of HAI cases, the severity of illness was comparable between MDR pathogen-associated HAI (MDR-HAI) and non-MDR pathogen-associated HAI cases (non-MDR HAI) (Table 2). In addition, almost all patient demographics, underlying chronic comorbidities, and clinical features were also comparable between these two groups. However, MDR-HAI cases were significantly more likely to be the second or third episode of nosocomial infections of the patients and occurred in later days of life than the non-MDR HAI cases (both *p* < 0.001). Therefore, MDR-HAI cases were more likely to occur in neonates with BPD (*p* = 0.054) and often occurred when the patients were on antibiotic treatment (*p* < 0.001). In addition, MDR pathogens accounted for 72.4% (*n* = 21) of the intra-abdominal infections. Most cases (219/275; 79.6%) had underlying chronic comorbidities, and 36.7% (*n* = 101) had two or more than two chronic comorbidities.

### 3.3. Therapeutic Outcomes and Impacts of Inappropriate Initial Antibiotics

Empiric antibiotics were prescribed in all these HAI cases, and 76.7% of the prescribed empiric antibiotics were broad-spectrum antibiotics, namely vancomycin or teicoplanin plus carbapenem or ceftazidime or cefotaxime (Table 3). Of all these HAI cases, 20.4% (*n* = 56) occurred while the patients were on antibiotic treatment. Inappropriate initial antibiotic treatment was significantly more often prescribed in the MDR-HAI group than to the non-MDR HAI group (51.0% versus 4.7%; *p* < 0.001). In addition, the modification of therapeutic antibiotics after antimicrobial susceptibility testing results was significantly more frequent in the MDR-HAI group than in the non-MDR HAI group; some of which were due to the poor control of the infectious focus. The median (IQR) treatment duration of all HAI cases was 11.0 (7.0–16.0) days. A total of 27 patients died within 72 h after the onset of HAIs due to fulminant or deteriorated courses, although all had finally received effective antibiotic treatment. Moreover, 42.9% (*n* = 118) had delayed control of the infectious focus, and the proportion was significantly higher in MDR-HAIs than in non-MDR HAIs (52.6% vs. 37.8%, respectively; *p* = 0.021). The HAI-attributable mortality rate and final in-hospital mortality rate of this cohort were 21.8% (*n* = 60) and 37.1% (*n* = 102), respectively. MDR-HAI cases were more likely to be treated with broad-spectrum antibiotics and for longer durations (15.0 ± 4.9 vs. 11.2 ± 3.3 days, respectively; *p* < 0.001), although the mortality rate was comparable with that of the non-MDR HAI cases.

The results of univariate and multivariate analyses of factors potentially associated with sepsis-attributable mortality in neonates with HAIs are summarized in Table 4. Lower gestational age and birth body weight were not independently associated with sepsis-attributable mortality or final in-hospital mortality. HAI-attributable mortality was not independently associated with antibiotic-resistant pathogens, inappropriate initial antibiotic treatment, or any specific pathogens. However, polymicrobial HAIs had a relatively higher sepsis-attributable mortality rate than monomicrobial HAIs. After adjustment, independent risk factors for the sepsis-attributable mortality of HAIs in neonates with severe respiratory failure were the presence of septic shock (OR: 3.61; 95% CI: 1.54–8.46, *p* = 0.003), higher illness severity (OR: 1.33; 95% CI: 1.04–1.72; *p* = 0.026 for every 3 increases in NTISS score), and patients with bronchopulmonary dysplasia (OR: 2.99; 95% CI: 1.47–6.09; *p* = 0.003). Good agreement between the observed and predicted values of the model (*p* = 0.66) was confirmed by the goodness-of-fit test of Hosmer and Lemeshow.

## 4. Discussion

Given the widespread use of broad-spectrum antibiotics in NICUs, the emergence of MDR pathogens is already a significant issue worldwide and has been the research focus in recent years [21,29,30]. MDR HAIs are often associated with more therapeutic challenges, higher rates of mortality and morbidity, and delayed appropriate antibiotic therapy [27,29,30,31,32]. The situation has increased complexity if neonates have multiple comorbidities and a higher severity of illness. To the best of our knowledge, this study is the first to investigate the clinical features and treatment outcomes of MDR HAIs in neonates with severe respiratory failure. We found that MDR pathogens account for approximately one-third of all neonatal HAIs when these patients have severe respiratory failure. The overall mortality was higher than the commonly reported mortality rate of 6–19% in neonates with late-onset sepsis [3,22,33]. After multivariate logistic regression, neonates with underlying bronchopulmonary dysplasia, a higher severity of illness, and the presence of septic shock were independently associated with a higher risk of final mortality.

Generally, MDR pathogens account for 18–35% of all neonatal sepsis in NICUs [12,30,31,32]. Our previous data showed that MDR Gram-negative bacteremia is associated with higher rates of infectious complications and overall case fatality [12], and other studies have found that the effective coverage rate of initial empiric therapy for MDR BSI is considerably lower than that for non-MDR BSI [12,30]. The worse outcomes of MDR bacteremia may result from delayed effective antibiotic therapy or sometimes from hypervirulent bacterial pathogens, such as MRSA or *Pseudomonas* species. Therefore, the early identification of neonates at high risk of MDR bacterial sepsis is very important, not only to prompt early effective therapy to optimize outcomes but also to avoid unnecessary use of broad-spectrum antibiotics [34,35]. In this cohort, broad-spectrum antibiotics were frequently prescribed because these patients had a high illness severity and the attending physicians could not take the risk of clinical deterioration. Since the emergence of MDR pathogens has been a significant issue in NICUs in recent years, we suggest that more surveillance and systemic data on regional epidemiology may be needed to develop optimized therapeutic strategies for neonatal severe sepsis and decrease the overuse of broad-spectrum antibiotics [36].

It has been estimated that approximately 50–88% of bacterial isolates from general NICUs are resistant to first-line antibiotics—ampicillin and gentamicin [37]. In our neonates with severe sepsis where MDR pathogens were suspected to be more prevalent, broad-spectrum antibiotics were frequently used at the onset of sepsis and initial inadequate antibiotic therapy was less common than we have thought. In this cohort, the underling BPD and higher severity of illness had a significantly greater impact on the outcomes than the initial timely administration of adequate antibiotics, which may have been due to the low percentage of only 22.9% of all HAI cases treated with inappropriate initial antibiotic therapy. Therefore, the influences of MDR pathogens may be masked by empirical broad-spectrum antibiotic therapy. However, the overuse of broad-spectrum antibiotics can potentially contribute to the emergence of MDR pathogens; therefore, antimicrobial stewardship programs are urgently needed and must be implemented to rationalize antibiotic use [37,38].

In our series, the severe respiratory failure of these neonates mainly came from underlying cardiopulmonary diseases, and less than one-fourth of these cases resulted from HAIs. The presence of underlying chronic comorbidities, prolonged intubation, artificial devices, and previous exposure to antibiotics all contributed to an increased risk of HAIs in NICUs [12,38]. Because this study focused on severe sepsis in neonates with respiratory failure, the pathogen distribution was notably different from that of general epidemiology in NICUs. In other words, high-risk microorganisms, including *Pseudomonas aeruginosa*-, MRSA-, *Stenotrophomonas*-, and ESBL-producing Gram-negative bacilli accounted for nearly one-third of all pathogens. A total of 46.5% of these HAI cases were caused by more than one microorganism, and positive bacterial culture from multiple sterile sites was also very common. In addition, nearly two-thirds of our subjects were extremely preterm neonates (GA < 28 weeks) and had multiple chronic comorbidities. All these factors could explain the significantly higher mortality rate in this cohort when compared with previous studies that reported a mortality rate of 11–19% in severe neonatal sepsis [3].

The definition of severe respiratory failure in this study was almost compatible with the criteria to use inhaled nitric oxide (iNO) [39,40]. A total of 24.7% of our cases received iNO during severe respiratory failure, although iNO has not been officially approved in extremely preterm neonates [41]. Because the majority of our neonates were very low birth-weight infants (birth body weight < 1500 g), extracorporeal membrane oxygenation was not available in our institute, and iNO in HFOV with maximal oxygen support may be the final treatment [42,43]. Therefore, a significant proportion of neonates died of cardiopulmonary failure, even though effective antibiotics had been prescribed and their infectious focuses were under control. We also found that neonates with severe BPD and secondary pulmonary hypertension had the highest rate of final mortality.

There were some limitations in this study. This was a retrospective study, and all patients were from a single center; therefore, applications of our study results may be less generalizable to other institutes. In critically ill neonates with severe respiratory failure, infections are often complicated with underling chronic comorbidities, and it may be difficult to classify therapeutic responses. Because polymicrobial microorganisms were often isolated from multiple sterile sites, it may be difficult to distinguish true pathogens from colonized microorganisms. In addition, this was not a randomized controlled trial, and most critically ill neonates received broad-spectrum antibiotics, especially those with multiple chronic comorbidities and high risk factors, which limited the significant influence of initial inappropriate antibiotics. However, the application of the most widely recognized diagnostic criteria and the complete follow-up of all our subjects were the strengths of our study.

## 5. Conclusion

In conclusion, MDR pathogens have emerged as a significant issue and account for approximately one-third of all HAI cases, especially in critically ill neonates with severe respiratory failure. Though we could not document the significant difference between MDR-HAIs and non-MDR HAIs in this cohort, neonates with MDR-HAIs had a longer duration of antibiotic treatment and a delayed resolution of clinical symptoms. We found that underlying chronic comorbidities and the severity of respiratory failure more significantly affected the outcomes because most patients received broad-spectrum antibiotics. However, the overuse of broad-spectrum antibiotics is now a significant issue; therefore, antibiotic stewardship programs for neonatal nosocomial infections are urgently needed.

## Figures and Tables

**Table 1 antibiotics-10-00459-t001:** Pathogen distribution of healthcare-associated infections (HAIs) in neonates with severe respiratory failure.

Pathogens	All HAI Episodes (Total *n* = 275)	Multidrug-Resistant Pathogens Associated HAIs (Total *n* = 95)	HAIs with Positive Cultures from Multiple Sterile Sites * (Total *n* = 141)
Gram-positive cocci	53 (19.3)	11 (11.6)	16 (11.3)
*Coagulase-negative Staphylococcus*	20 (7.3)	0 (0)	7 (5.0)
*Methicillin-resistant Staphylococcus aureus*	11 (4.0)	11 (11.6)	4 (2.8)
*Methicillin-sensitive Staphylococcus aureus*	12 (4.4)	0 (0)	3 (2.1)
*Enterococcus* spp.	3 (1.1)	0 (0)	0 (0)
*Group B Streptococcus*	3 (1.1)	0 (0)	0 (0)
*Listeria monocytogenes*	4 (1.5)	0 (0)	2 (1.4)
Gram-negative bacilli	94 (33.5)	23 (24.2)	27 (19.1)
*Escherichia coli*	32 (11.6)	7 (7.4)	11 (7.8)
*Klebsiella spp.*	12 (4.4)	4 (4.2)	4 (2.8)
*Enterobacter spp.*	6 (2.2)	2 (2.1)	0 (0)
*Serratia marcescens*	9 (3.3)	0 (0)	1 (0.7)
*Acinetobacter baumannii*	12 (6.1)	0 (0)	5 (3.5)
*Pseudomonas aeruginosa*	10 (4.4)	4 (4.2)	4 (2.8)
*Stenotrophomonas maltophilia*	3 (1.1)	3 (3.2)	1 (0.7)
Others **	10 (3.6)	3 (3.2)	1 (0.7)
Polymicrobial microorganisms	128 (46.5)	61 (64.2)	98 (69.5)
Two Gram-positive cocci (GPC)	9 (3.3)	2 (2.1)	5 (3.5)
Two Gram-negative bacilli (GNB)	16 (5.8)	7 (7.4)	15 (10.6)
Combined GPC and GNB	34 (12.4)	10 (10.5)	27 (19.1)
≥3 microorganisms	50 (18.2)	33 (34.7)	43 (30.5)
Any combination with fungi species	19 (6.9)	9 (9.5)	8 (5.7)

* Indicates positive culture isolated from two or more than two otherwise sterile sites, which include blood, cerebrospinal fluid, pleural effusion, ascites, urine, catheter tip (excluding contamination), and tracheal aspirates (excluding colonization); ** Including *Corynebacterium striatum* (3), *Morganella* species (2), *Citrobacter koseri* (2), *Moraxella catarrhalis* (1), and *Burkholderia cepacia* (2).

**Table 2 antibiotics-10-00459-t002:** Patient demographics, characteristics, and clinical presentation of all neonatal healthcare-associated infections (HAIs) in Chang Gung Memorial Hospital (CGMH) from January 2014 to May 2020.

Characteristics	All HAI Episodes (Total *n* = 275)	MDR Pathogen-Associated HAI Episodes (Total *n* = 95)	Non-MDR Pathogen-Associated HAI Episodes (Total *n* = 180)	*p*-Values
Cases demographics				
Gestational age (weeks), median (IQR)	26.0 (25.0–29.0)	26.0 (25.0–28.0)	26.0 (25.0–29.0)	0.277
Birth weight (g), median (IQR)	838.0 (721.0–1080.0)	793.0 (721.0–1010.0)	860 (712.8–1220.0)	0.104
Gender (male/female), *n* (%)	172 (62.5)/102 (37.5)	61 (64.2)/34 (35.8)	112 (62.2)/68 (37.8)	0.794
5 min Apgar score ≤ 7, *n* (%)	148 (53.8)	49 (51.6)	99 (55.0)	0.771
Inborn/outborn, *n* (%)	224 (81.5)/51 (18.5)	77 (81.1)/18 (18.9)	147 (81.7)/33 (18.3)	0.901
Birth by NSD/cesarean section, *n* (%)	109 (39.6)/166 (60.4)	32 (33.7)/63 (66.3)	77 (42.8)/103 (57.2)	0.155
Respiratory distress syndrome (≥Gr II), *n* (%)	190 (69.1)	69 (72.6)	121 (67.2)	0.411
Intraventricular hemorrhage (≥Stage III), *n* (%)	31 (11.3)	11 (11.6)	20 (11.1)	0.907
Underlying Chronic Comorbidities, *n* (%)				
Neurological sequelae	75 (27.3)	29 (30.5)	46 (25.6)	0.395
Bronchopulmonary dysplasia	190 (69.1)	73 (76.8)	117 (65.0)	0.054
Complicated cardiovascular diseases	22 (8.0)	12 (12.6)	10 (5.6)	0.059
Symptomatic patent ductus arteriosus	80 (29.1)	24 (25.3)	56 (31.1)	0.332
Gastrointestinal sequelae	28 (10.2)	13 (13.7)	15 (8.3)	0.208
Renal disorders	6 (2.2)	2 (2.1)	4 (2.2)	0.950
Congenital anomalies	18 (6.5)	8 (8.4)	10 (5.6)	0.443
Presences of any chronic comorbidities	219 (79.6)	83 (87.4)	136 (75.6)	0.027
Presences of more than one comorbidities	101 (36.7)	37 (38.9)	64 (35.6)	0.600
Day of life at onset of HAIs (day), median (IQR)	25.0 (12.0–56.0)	44.0 (21.0–73.0)	22.0 (8.0–42.0)	<0.001
On antibiotic treatment at onset of HAIs, *n* (%)	56 (20.4)	31 (32.6)	25 (13.9)	<0.001
Use of TPN and/or intrafat, *n* (%)	177 (64.4)	62 (65.6)	115 (63.9)	0.895
Use of central venous catheter, *n* (%)	265 (96.4)	93 (97.9)	172 (95.6)	0.893
Infectious focus, *n* (%)				<0.001
Bloodstream infection only	113 (41.1)	31 (32.6)	82 (45.6)	
Ventilator-associated pneumonia (VAP)	88 (32.0)	30 (31.6)	58 (32.2)	
Catheter-related bloodstream infection	35 (12.7)	11 (11.6)	24 (13.3)	
Urinary tract infection	6 (2.2)	2 (2.1)	4 (2.2)	
Intra-abdominal infection	16 (5.8)	13 (13.7)	3 (1.7)	<0.001
Meningitis	4 (1.5)	0 (0)	4 (2.2)	
VAP plus intra-abdominal infection	13 (4.7)	8 (8.4)	5 (2.8)	
Clinical features, *n* (%)				
On HFOV/conventional ventilator	158 (57.5)/117 (42.5)	57 (60.0)/38 (40.0)	101 (56.1)/79 (43.9)	0.608
On inhaled nitric oxide (iNO)	68 (24.7)	26 (27.4)	42 (23.3)	0.466
Oxygenation index at onset of bacterial sepsis ^#^	14.0 (8.0–34.0)	14.0 (8.0–30.0)	15.5 (7.3–35.8)	0.990
Septic shock	175 (63.6)	55 (57.9)	120 (66.7)	0.187
Metabolic acidosis	177 (64.4)	61 (64.2)	116 (64.4)	0.969
Coagulopathy	186 (67.6)	67 (70.5)	119 (66.1)	0.500
NTISS score at onset of HAI, median (IQR)	27.0 (23.8–29.0)	27.8 (23.5–29.5)	27.0 (24.0–28.8)	0.373
Presences of bacteremia	252 (91.6)	89 (93.7)	163 (90.1)	0.494
Requirement of blood transfusion *	216 (78.5)	78 (82.1)	138 (76.7)	0.355

NSD: normal spontaneous delivery; IQR: interquartile range; HFOV: high-frequency oscillatory ventilator; NTISS score: Neonatal Therapeutic Intervention Scoring System; TPN: total parenteral nutrition.; * Including leukocyte poor red blood cell and/or platelet transfusion; ^#^ Data are median (interquartile range).

**Table 3 antibiotics-10-00459-t003:** Therapeutic intervention and outcomes of all neonatal healthcare-associated infections (HAIs) in neonates with severe respiratory failure in the CGMH from January 2014 to May 2020.

Characteristics	All HAI Episodes (Total *n* = 275)	MDR Pathogen-Associated HAI Episodes (Total *n* = 95)	Non-MDR Pathogen-Associated HAI Episodes (Total *n* = 180)	*p* Values
Therapeutic intervention, *n* (%)				
Initial empiric antibiotics				
Inappropriate initial antibiotics	54 (22.9)	45 (51.0)	9 (4.7)	<0.001
Use of first line antibiotics	64 (23.3)	19 (20.0)	45 (25.0)	0.372
Use of broad-spectrum antibiotics	211 (76.7)	76 (80.0)	135 (75.0)	0.372
Modification of therapeutic antibiotics	144 (52.4)	59 (62.1)	85 (47.2)	0.022
Therapeutic antibiotics				
Use of first line antibiotics	50 (18.2)	10 (10.5)	40 (22.2)	0.021
Use of broad-spectrum antibiotics	225 (81.8)	85 (89.5)	140 (77.8)	0.021
Duration of antibiotic treatment (day), mean ± SD	12.6 ± 3.8	15.0 ± 4.9	11.2 ± 3.3	0.034
Therapeutic outcomes, *n* (%)				
Failure to control infectious focus in 1 week	118 (42.9)	50 (52.6)	68 (37.8)	0.021
Duration of mechanical ventilation	62.0 (26.0–91.0)	72.0 (36.0–102.0)	55.5 (16.0–84.0)	0.011
Duration of hospitalization, day (median (IQR))	88.0 (44.0–130.0)	86.0 (62.0–132.0)	88.5 (22.8–128.3)	0.246
Sepsis-attributable mortality	60 (21.8)	18 (18.9)	42 (23.3)	0.445
Final in-hospital mortality	102 (37.1)	34 (35.8)	68 (37.8)	0.794

IQR: interquartile range.

**Table 4 antibiotics-10-00459-t004:** Multivariate logistic regression analysis for independent risk factors of sepsis-attributable mortality in neonates with healthcare-associated infections.

Variables	Univariate Analysis		Multivariate Analysis	
	OR (95% CI)	*p* Values	Adjusted OR (95% CI)	*p*-Values
Gestational age				
<26 weeks	1.49 (0.51–4.33)	0.464		
26–28 weeks	0.73 (0.29–1.85)	0.507		
29–33 weeks	0.90 (0.36–2.27)	0.820		
≥34 weeks	1 (reference)			
Septic shock	4.86 (2.20–10.73)	<0.001	3.61 (1.54–8.46)	0.003
On HFOV vs. conventional ventilator	1.99 (1.08–3.68)	0.028	0.623 (0.26–1.50)	0.290
Inappropriate initial antibiotics	1.50 (0.76–2.97)	0.239		
MDR pathogen-associated HAIs	0.77 (0.41–1.43)	0.403		
Polymicrobial HAIs	1.89 (1.02–3.52)	0.045	1.22 (0.61–2.44)	0.569
Bronchopulmonary dysplasia	4.24 (2.24–8.03)	<0.001	2.99 (1.47–6.09)	0.003
Severity of illness at onset of HAIs				
Every 3 increase in NTISS scores	1.37 (1.13–1.64)	0.001	1.33 (1.04–1.72)	0.026
Thrombocytopenia	1.57 (0.82–3.01)	0.170		

HFOV: high frequency oscillatory ventilator; OR: odds ratio; 95% CI: 95% confidence interval; MDR: multidrug-resistant; NTISS: Neonatal Therapeutic Intervention Scoring System; HAI: healthcare-associated infection.

## Data Availability

The datasets used/or analyzed during the current study available from the corresponding author on reasonable request.

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
