# Peer review of "Multidrug-Resistant Healthcare-Associated Infections in Neonates with Severe Respiratory Failure and the Impacts of Inappropriate Initial Antibiotic Therap"

_antibiotics, 2021, doi:10.3390/antibiotics10040459_

Round 1

Reviewer 1 Report

Minor concerns:

Line 52:  Escherichia coli instead of E. coli; Streptococcus agalactiae (Group B strep [GBS]) instead of group B streptococcus

Line 57: correct: “or in critically neonates with severe respiratory failure”

Line 58: “In these situations” instead of “In this situations”

Line 63: correct: “these patients are likely to have antibiotic overuse”

Line 65: correct “or those present with critical illness is common”

Lines 73-75:  “All neonates with….  were retrieved from our data base for analysis”: their records were retrieved, not the neonates themselves, please correct accordingly.

Lines 83-86: Move the sentence “This study was approved by the Institutional 83 Review Board of CGMH (the certificate no. 201700988A3), and written informed consent was waived because all patient records/information were anonymized and de-identified prior to analysis” to the Acknowledgement section.

Line 87: change the heading “Definition” to something more descriptive to what is narrated in the following text”.

Line 89: What is meant by “observed”?  Please use a more descriptive term.

Line 90: Introduce VAP acronym here.

Lines 89-91: Substitute fulfilled with fulfilling in the sentence: “We retrospectively observed all radiological, clinical and microbiological data of neonatal VAP episodes and those fulfilled the strict diagnostic criteria of VAP in the NICU (appendix Table 1) were enrolled”

Lines 92-93: Reword the sentence: “Only the first episode of HAI in each patient who were under the situation of severe respiratory failure was enrolled for analyses”

Lines 93-94: Reword the sentence: “Severe respiratory failure in neonates is defined when the neonate had at least one of the following situations:” to “Severe neonatal respiratory failure is defined by:”

Line 95: Reword: “Requirement of fraction of inspirited oxygenation” to:  “Requirement of inspired oxygen fraction”.

Line 100:  “indicates” instead of “indicating” and insert “otherwise” before “sterile site” and combine with the following sentence.

Lines 102-103: Italicize scientific names of all bacteria listed.

Line 106: Please clarify how “all positive blood cultures were considered”? What was considered and how?

Lines 116-118: Please reword the last words of this sentence: “When one of the isolated bacterial strains in polymicrobial HAIs was resistant to ≥ 3 antimicrobial categories, it was also defined as the MDR HAI episode.”

Line 128:  Change “standard form” to “standard forms”.

Line 129: insert “presumed” before the word “sterile”.

Line 132: missing citation for the “standard definition”.

Lines 132-133: Please reword that sentence.

Line 144-146: Please reword that sentence.

Lines 158-160: Please use the word “cases” instead of “episodes” and correct the “sterile sites” as suggested previously.

Every mention please use lower case for gram positive or gram negative. Capitalized Gram is reserved when referring to the stain itself.

Lines 200-202: Remove “be” and add “to” after the word “than” from the sentence:  “Inappropriate initial antibiotic treatment was significantly more often be prescribed in MDR-HAIs than the non-MDR HAI group (51.0% versus 4.7%, P < 0.001).”

Line 204: correct the typo “gropu” to “group” and substitute “often” with “frequent”.

Line 206: substitute “cases” with “patients”.

Line 213: substitute “were” with “was”.

Line 278: “pathogen” instead of “the pathogens”

 Major concerns:

Please state the hypothesis driving this investigation the end of the introduction.

From lines 104-105: The “limited-spectrum antibiotic” definition under which ampicillin/sulbactam, oxacillin, ceftriaxone and gentamicin are classified by the authors is difficult to understand.  Perhaps “limited spectrum” is a novel definition distinct from “broad” and “narrow” spectrum that requires clarification by the authors. 

From line 107:  This is important because it concerns the use of the word “inappropriate initial antibiotic treatment” a foundational parameter of this study: Were bacterial cultures tested for resistance?  Were patient cultures isolated?  If so how? By MIC/MBCS?  By Disk diffusion? If not, how could isolated bacterial strains be defined as resistant?

Lines 107-108: “Inappropriate initial antibiotic treatment was considered when one of the isolated bacteria strains in the HAI episodes was resistant to the empirical treatment.” The word “considered” does not sufficiently define how this information was processed.  Furthermore, empirical treatment is inferred to mean that if a patient’s condition did not improve upon treatment with a given preventative antibiotic, then the bacterial strain (sic) was resistant to that antibiotic.  However, this is a fallacious inference as sub-culturing of the “bacterial strain” and testing against antibiotics in vitro would be necessary to draw such a conclusion.

Paragraph from line 104 through 118 is confusing because it does not follow a logical progression; hence my previous comments. The authors need to first of all clarify what data specifically they had available from the database for analysis before anything else.  Explain that cultures of bacteria were taken from neonates, cultured axenically, how they were typed and disk diffusion assays performed in duplicates/triplicates or what may be the case. 

Would be informative to know how many total neonates were in care of the health care facility during the 1/2014-5/2020 window of time of this analysis to understand the overall impact of Hospital Acquired Infections at this site.

Lines 158-160: the authors claim that 91.6% of HAI neonates had positive blood cultures and 51.3% had positive cultures from other locations (evidently sites that are not sterile).  However, at face value, taken alone, this information is not informative because there are no comparisons (i.e. controls).  To attribute any significance to these findings the authors would have to compare the rates of positivity of this population to, for example, cohorts not diagnosed with HAI which would function as a negative control population.  The same applies for the results of low-birth weight.

Lines 165-171: Certainly S. aureus and P. aeruginosa can be opportunistic HAI pathogens, but because no comparison with other, non-HAI neonate cohorts is presented, it is impossible to ascertain the significance of this finding. 

There is no reason to hypothesize that disease gravity or demographics would be impacted by the presence of MDR pathogens.  As a result, Table 2 contains virtually exclusively negative data; I suggest removing all criteria that do not yield a significant statistical difference between groups.

Author Response

Dear reviewer,

Besides, I have completed English editing from professional English editing company, thank you

Best regard,

Tsai Ming Horng

Reviewer 2 Report

This is a very interesting study of 275 critically ill very preterm neonates with severe respiratory failure who had HAIs. Ninety-five episodes (34.5%) were caused by MDR 25 pathogens, and 141 (51.3%) episodes had positive bacterial cultures from multiple sterile sites. In 26 this cohort, the MDR-HAI group was more likely to receive inappropriate initial antibiotic therapy.

Empirically broad-spectrum antibiotics were prescribed in 76.7% of cases, hoewever inappropriate initial antibiotic treatment was not significantly associated with worse outcomes. Independent risk factors for sepsis-attributable mortality in neonates with severe respiratory failure included the presence of septic shock, higher illness severity, and neonates with bronchopulmonary dysplasia. It is very worrying that the percentage of MDR bacteria is very high in these very immature infants who require prolonged hospitalization in NICU. Thiσ underlie the importance of a strict antibiotic policy in the unit based on the strains that are isolated from the patients. It is a very interesting study and I suggest that it gets published in Antiβiotics

This is a single center retrospective study of all episodes of HAIs in neonates with severe respiratory failure in a tertiary level NICU. All clinical features, microbiology, therapeutic interventions and outcomes were compared between the MDR-HAIs and non- 22 MDR HAIs groups. Multivariate regression analyses were used to investigate independent risk fac tors for sepsis-attributable mortality 

A total of 275 critically ill neonates with severe respiratory failure who had HAIs were enrolled. Ninety-five episodes (34.5%) were caused by MDR pathogens, and 141 (51.3%) episodes had positive bacterial cultures from multiple sterile sites. In this cohort, the MDR-HAI group was more likely to receive inappropriate initial antibiotic therapy 

Empirically broad-spectrum antibiotics were prescribed in 76.7% of cases, and inappropriate initial antibiotic treatment was not significantly associated with worse outcomes. Independent risk factors for sepsis-attributable mortality in neonates with severe respiratory failure included presence of septic shock, higher illness severity, and neonates with bronchopulmonary dysplasia 

All neonatal early-onset sepsis, late-onset sepsis, meningitis and sterile sites culture positive nosocomial infections were enrolled, which included ventilator-associated pneumonia, catheter-related BSIs, and intra-abdominal infections Inappropriate initial antibiotic treatment was considered when one of the isolated bacteria strains in the HAI episodes was resistant to the empirical treatment 

Page 4 define VAP 

The authors state that MDR-HAI episodes were more likely to be treated with broad-spectrum antibiotics and for longer duration although the mortality rate were comparable with the non-MDR HAI episodes. 

HAI-attributable mortality was not independently associated with antibiotic-resistant pathogens, inappropriate initial antibiotic treatment, or any specific pathogens 

However the authors state later in the paper that after multivariate logistic regression, neonates with underlying neurological sequelae and delayed initial appropriate antibiotics were independently associated with higher risk of final mortality and this issue needs to be clarified 

Therefore, early identification of neonates at high risk of MDR bacterial sepsis is very important, not only to prompt early effective therapy to optimize outcomes but also avoid unnecessary use of broad-spectrum antibiotics 

The authors need to emphasizemore that due to multiresistant organisms in the extremely low birth weight infants antimicrobial stewardship programs remain necessary and must be implemented to rationalize antibiotic use and avoid overuse of antibiotics. 

Also, the authors might include in their paper an algorithm/protocol to explain in more detail how they escalate antibiotic use and the way empirical antibiotics are applied in their patients based on their condition and length of stay in the NICU. 

Author Response

Dear reviewer:

      Please see the attachment. Besides, I have completed the English editing from professional English editing company, thank you.

Best regard,

Tsai Ming Horng

Round 2

Reviewer 1 Report

No comments, thank you.